# Application of AI Mind Mapping in Mental Health Care

**DOI:** 10.3390/healthcare13151885

**Published:** 2025-08-01

**Authors:** Hsin-Shu Huang, Bih-O Lee, Chin-Ming Liu

**Affiliations:** 1Department of Nursing, Central Taiwan University of Science and Technology, Taichung 40601, Taiwan; 2College of Nursing, Kaohsiung Medical University, Kaohsiung 80708, Taiwan; biholee@kmu.edu.tw; 3Department of Psychiatry, Cheng Ching Hospital, Taichung 40045, Taiwan; 4841@ccgh.com.tw

**Keywords:** artificial intelligence, empowerment, thinking function

## Abstract

Background: Schizophrenia affects patients’ organizational thinking, as well as their ability to identify problems. The main objective of this study was to explore healthcare consultants’ application of AI mind maps to educate patients with schizophrenia regarding their perceptions of family function, social support, quality of life, and loneliness, and to help these patients think more organizationally and understand problems more effectively. Methods: The study used a survey research design and purposive sampling method to recruit 66 participants with schizophrenia who attended the psychiatric outpatient clinic of a hospital in central Taiwan. They needed to be literate, able to respond to the topic, and over 18 years old (inclusive), and they attended individual and group health education using AI mind maps over a 3-month period during regular outpatient clinic visits. Results: The study results show that patients’ family function directly affects their quality of life (*p* < 0.05) and loneliness (*p* < 0.05), satisfaction with social support affects quality of life and loneliness directly (*p* < 0.05), and satisfaction with social support is a mediating factor between family function and quality of life (*p* < 0.05), as well as a mediating factor between family function and loneliness (*p* < 0.05). Conclusions: Therefore, this study confirms the need to provide holistic, integrated mental health social care support for patients with schizophrenia, showing that healthcare consultants can apply AI mind maps to empower patients with schizophrenia to think more effectively about how to mobilize their social supports.

## 1. Introduction

The World Health Organization’s 2001 report, “Mental health: new understanding, new hope”, sets out three directions for the treatment of people with mental disorders and psychosocial disabilities: (1) there must be no reason to discriminate against people with mental disorders and psychosocial disabilities; (2) the right of persons with mental disorders and psychosocial disabilities to receive treatment and care in their own communities must not be exploited; and (3) people with mental disorders and psychosocial disabilities have the right to receive treatment in the least restrictive environment and with the least restrictive or invasive treatment [1]. Social reintegration for individuals with schizophrenia is a critical aspect of their recovery and overall well-being. This process involves helping individuals with schizophrenia return to their communities and engage in social, educational, and occupational activities.

In 2022, the WHO stated that communities need to have a better understanding of the concept of mental health. For the treatment of people with mental disorders and psychosocial disabilities, communities should provide health services; in particular, community mental services need to cover mental health centers and teams, psychosocial rehabilitation, support system establishment, and living assistance. Influenced by Western initiatives, Taiwan has promoted the concepts of deinstitutionalization and community care. According to policy changes, people with mental disabilities have gradually transitioned from placement and long-term hospitalization to a community mental healthcare model [2].

Patients with schizophrenia face many challenges in daily life, including emotional management, social interaction, and self-expression. As a visual thinking tool, mind maps can play an auxiliary function in the social life of patients with schizophrenia.

The application of AI mind maps can help these patients organize their thoughts, express their emotions, and conduct self-reflection more effectively. These tools not only provide a way to think visually but also promote problem-solving.

The main objective of this purposive sampling survey study was to explore healthcare consultants’ application of AI mind maps to educate patients with schizophrenia in their perceptions of family function, social support, quality of life, and loneliness, and to help these patients think more organizationally and understand problems more effectively.

An individual’s family can provide support to reduce stress in their life [3], comprising an important factor affecting mental health and prognosis. If there is a person with schizophrenia in the family, the entire family will face challenges. The better the family functions, the more efficient it is in problem-solving [4]. Zhou’s (2014) research results showed that good family function can have positive effects on the mental health of both patients and their families, as well as improving family satisfaction [5].

Social support includes interpersonal communication and the provision of various resources, such as information, substantial help, and psychological support. Communication is the basis of social support [6,7]. Patients with schizophrenia often face difficulties in developing interpersonal relationships and maintaining social relationships [8,9,10]. The longer the illness lasts, the easier it is for the individual to be discriminated against by society, and the harder it is for them to receive social support aside from psychiatric treatment [11,12,13]. As social support decreases, the risk of disease relapse increases and patients become increasingly disconnected from society, finally becoming isolated and marginalized [13].

In the treatment of patients with schizophrenia, emphasis is placed on symptom relief and functional recovery, which are important indicators for medical practitioners to use in evaluating whether patients have recovered [14]. According to past research, patients with schizophrenia have significantly lower physical, mental, social, and environmental health, as well as a lower quality of life, than non-schizophrenic people. This may be attributed to mental symptoms, social isolation, stigma and discrimination, cognitive impairment, and low employment opportunities [15]. The above situation is a common problem that patients with schizophrenia encounter in their daily lives.

At present, medical treatments for patients with schizophrenia have begun to shift from traditional symptom reduction to focusing on quality of life and striving to improve overall mental symptoms [16].

Loneliness is a common phenomenon for patients with schizophrenia, but the literature suggests that they have never received help for it [17,18], due to a lack of opportunities and limited participation in social activities and interactions with others in daily life [19].

Scholars have concluded that the reasons why patients with schizophrenia may experience loneliness are as follows: (1) patients with schizophrenia have limited access to social networks; (2) patients with schizophrenia are severely stigmatized by and excluded from society; (3) they tend to deliberately keep a distance in their interactions with others due to the impacts of their symptoms; (4) patients with schizophrenia have limited financial and other resources to maintain social relationships; (5) they participate less in social activities or have fewer good social experiences; (6) they have internalized social stigma and low self-esteem [20,21,22,23]. Due to barriers preventing communication with society, patients with schizophrenia often feel isolated and lack good relationships. However, during the treatment process, limited attention is paid to the loneliness caused by the lack of social networks [24], which leads these patients to become increasingly separated from society and, eventually, live alone.

In the modern education and work environment, mind mapping as a visualization tool has an auxiliary function for patients with schizophrenia [25]. The emergence of AI mind mapping has further enhanced the functions of mind mapping and has been applied to help patients with schizophrenia to organize and process information more effectively (Figure 1 and Figure 2).

Improve information organization capabilities

The non-linear structure of mind maps allows users to organize their thoughts in a more natural way. This is especially important for patients with schizophrenia, who may have difficulties with traditional linear learning and information processing. AI mind mapping allows for the automatic generation of structured graphics to help users simplify complex information into an easy-to-understand visual format.

2.Enhance memory and recall ability

The query references research indicating that approximately 65% of the population are visual learners and that mind maps are particularly effective for this group, especially as described in "The Mind Map Book" (ISBN 978-0-452-27322-1) [25]. AI mind maps use colors, images, and keywords to strengthen memory, which is especially beneficial for patients with schizophrenia, who have short-term memory impairment.

3.Reduce anxiety and stress

The visual nature of mind maps can help patients with schizophrenia to reduce anxiety during study and work. By presenting information graphically, users can more clearly see the structure of a task, thereby reducing the stress caused by information overload.

4.Promote creativity and mental flexibility

AI mind maps not only comprise basic graphic structures but also provide creative suggestions to help users explore new ideas and connections. This function can stimulate and promote deeper thinking for patients with schizophrenia, who need flexible thinking.

5.Support collaboration and social interaction

Many AI mind maps support multi-person collaboration, which is crucial for patients with schizophrenia participating in a team. These tools promote social interaction, help patients with schizophrenia to express their ideas in group discussions, and enhance their social skills.

Monica AI is a multi-functional artificial intelligence assistant that is particularly suitable for generating mind maps, which is of great significance for the education of patients with schizophrenia.

The following are its specific applications in this field:Visual thinking and learning

Monica AI mind mapping function can transform complex concepts into intuitive graphics, which can help patients with schizophrenia better understand and remember information. By presenting information graphically, users can more easily grasp key concepts and connections, thereby improving learning outcomes.

2.Automatic generation of mind maps

The healthcare consultant only needs to enter text or upload relevant information, and Monica AI can automatically generate a mind map. This function is particularly suitable for situations where ideas need to be organized quickly. For example, in individual and group health education, outlines or key points can be quickly generated to help patients with schizophrenia more effectively follow the teaching progress.

3.Enhancing participation and interactivity

In health education activities, Monica AI can be used as an auxiliary tool to encourage patients with schizophrenia to participate in the learning process. Through interactive mind map generation, users can add their own thoughts and questions to the map, which not only enhances their sense of participation but also promotes their depth of thinking.

4.Supporting multiple input formats

Monica AI supports a variety of input formats, including PDFs, Word documents, and YouTube videos. This feature is particularly important for patients with schizophrenia, who need a variety of learning materials and can be taught by healthcare consultants to obtain information from different media and develop a more comprehensive understanding.

5.Improving learning efficiency

By using the mind maps generated by Monica AI, patients with schizophrenia can master content faster, which is crucial to improving their learning efficiency. The structured nature of mind maps makes the absorption and review of information more efficient, helping them to better apply what they have learned in their daily lives.

The Monica AI mind mapping tool has wide application potential in the health education of patients with schizophrenia. It can effectively improve learning effects, enhance participation, and promote the understanding and memorization of information.

In summary, schizophrenia affects patients’ organizational thinking and ability to identify problems. AI mind maps play multiple roles in supporting patients with schizophrenia when studying or at work, from improving information organization capabilities to enhancing their thinking and helping to solve problems.

## 2. Materials and Methods

### 2.1. Study Design and Participants

This study used a survey research design and purposive sampling method to recruit participants and obtain their consent to participate. All questionnaires were filled out verbally and anonymously. The questionnaire content was only used for research and analysis. Ethical regulations were strictly adhered to, and the privacy of the participants was protected.

The participants of this study were patients with schizophrenia who attended the psychiatric outpatient clinic of a hospital in central Taiwan. To be eligible for the study, participants must have been assessed by their psychiatrist as having stable mental symptoms. They also needed to be literate, able to respond to the topic, and over 18 years old (inclusive). Patients attended individual and group health education using AI mind maps over a 3-month period during regular outpatient clinic visits.

### 2.2. Measurement

#### 2.2.1. Demographic Variable Questionnaire

The background information of the participants was collected and included gender, age, education level, and religious beliefs.

#### 2.2.2. Family Function Scale

The family function scale was used to understand whether the family is a resource or a source of pressure for the participants. This scale’s content is divided into five levels: adaptability, cooperation, growth, emotion, and intimacy. It measures the ability to use family resources and jointly make decisions, responsibility cultivation ability, whether the family can support one’s self-achievement ability, interactive relationship ability, and the ability to get along with family members. There are five questions in this scale, which are evaluated using a three-point scoring method ranging from 0 to 2, with a total score of 10 points. A total score of 0–3 is considered severely difficult, 4–6 is considered mildly difficult, and 7–10 indicates good function, meaning that patients have a sufficient ability to face problems. Smilkstein et al. (1982) [26] tested the reliability of the scale in terms of internal consistency; the Cronbach’s alpha value was in the range of 0.80 to 0.86. In terms of validity, different studies have reported different values, with a range between 0.64 and 0.8, thus demonstrating a certain level of validity [26].

#### 2.2.3. Social Support Scale

The original social support scale was compiled by Sarason, Levine, Basham, and Sarason in 1983 [27,28]. The original scale had a total of 27 questions, which was later revised to 20 questions. This scale is divided into two parts: number of social supports (SSQ (N)) and social support satisfaction (SSQ (S)). The number of social support options ranges from 0 to 9 individuals; social support satisfaction is measured using a six-point Likert scale, with 1 indicating very dissatisfied and 6 indicating satisfied. This study adopted the Chinese social support scale translated by Wu Jingji in 1986 and revised by Zhang Zhiyuan in 1989. The scale’s Cronbach’s alpha value for the number of social supports (SSQ (N)) was 0.9269, and for social support satisfaction (SSQ (S)) it was 0.94, which shows good internal consistency. Its validity was tested using factor analysis, and the loadings of the number of significant social supports and social support satisfaction were greater than 0.40, indicating that the revised scale has good construct validity [28].

#### 2.2.4. Quality of Life Scale

The original quality of life scale, WHOQOL-100, had a total of 100 questions, which could be divided into four major categories. However, as the questionnaire was too long, WHOQOL-BREF was developed, with a total of 28 questions. The scale itself has 26 universal questions: 7 are related to physical health, 6 to mental health, 4 to social relations, and 9 to the environment. There are two additional questions asking the respondent for an overall evaluation of their quality of life.

This scale measures health and general quality of life in other aspects. It uses a five-point Likert scale. In the reliability and validity test, the Cronbach’s alpha for the entire questionnaire was 0.91, and the Cronbach’s alpha for the four major categories was in the range of 0.70–0.77 [29,30]. The content validity between a question and the level it belongs to was 0.57–0.91. The content validity between a category and overall quality of life scores was 0.64–0.85, with *p* < 0.01, indicating that the reliability and validity of this questionnaire are both good [31].

#### 2.2.5. Loneliness Scale

Russell et al. (1980) [32] developed the UCLA Loneliness Scale Version 3 to measure the level of loneliness using a four-point Likert scale, with the options being “never,” “rarely,” “often,” and “always;” there are 20 questions in total. It is a one-dimensional measurement tool. The internal consistency reliability of this scale was 0.89 to 0.94, and the test–retest reliability was 0.73, which shows that this scale has good reliability. The higher the final summed score, the more lonely a respondent feels [32,33].

### 2.3. Data Collection and Analysis

This study calculated the sample size using G*power version 3.1 statistical software, with an effect size of 0.15, an α error probability of 0.05, and an estimated sample loss rate of 20%. AI mind maps were used during regular outpatient clinic attendance for individual and group health education; 66 persons with schizophrenia agreed to participate in this study.

After the data were collected, decoded, logged, and archived, a descriptive and inferential statistical analysis was conducted using the SPSS 26.0 for Windows/PC software suite, and α = 0.05 was set as the standard for significant differences.

#### 2.3.1. Descriptive Statistics

The distribution of categorical variables is presented in terms of frequency distribution and percentage, and the distribution of continuous variables is presented in terms of number (n) and percentage (%).

#### 2.3.2. Inferential Statistics

A set of regression analyses was used to verify the causal mechanism. The unique effect of an independent variable on the dependent variable is called the direct effect, and the independent variable affects the dependent variable through the mediator, which is called the indirect effect. The sum of the two effects is the so-called total effect, which can be obtained using path analysis to discover the key to the true core of the dependent variable.

### 2.4. Ethical Considerations

This research project was reviewed and approved by the IRB of the China Medical University Hospital (No.: CRREC-111-052, date 16 March 2023). The participants had the right to withdraw from the study at any time and could ask for further information. The questionnaires were numbered anonymously to delink and ensure confidentiality. The names and conditions of the participants will never be publicized, and the results are for academic use only.

## 3. Results

### 3.1. Study Sample Characteristics

In terms of demographic characteristics, the proportion of the participants was 50% male (33 people) and 50% female (33 people). Approximately 55% (36 people) of participants were aged 18–40 years, and 45% (30 people) were aged 41 years or above (inclusive). A total of 30% (20 people) of participants reported a junior high school education or below, and 70% (46 people) reported a high school vocational education or above. Approximately 47% (31 people) of participants reported having religious beliefs, and 53% (35 people) reported having no religious beliefs.

### 3.2. Mediation Effect Analysis

#### 3.2.1. Mediating Effect of Family Function, Social Support Satisfaction, and Quality of Life

In exploring whether “social support satisfaction” is a mediating factor between “family function” and “quality of life” (as shown in Table 1 and Figure 3), it was found that “family function” has a statistically significant impact on “quality of life” (*p* < 0.05). “Family function” also has a statistically significant impact on “satisfaction with social support” (*p* < 0.05).

Then, the Bootstrap method was used to test whether “satisfaction with social support” is a mediating variable, and it was found that the direct effect is statistically significant (the confidence interval does not include 0), indicating that “satisfaction with social support” affects “quality of life” directly; indirectly, the mediating effect is statistically significant (the confidence interval does not include 0), indicating that “social support satisfaction” is the mediating factor between “family function” and “quality of life.”

#### 3.2.2. Mediating Effect of Family Function, Social Support Satisfaction, and Loneliness

When analyzing whether “social support satisfaction” is a mediating factor between “family function” and “loneliness” (as shown in Table 2 and Figure 4), it was found that “family function” has a statistically significant impact on “loneliness” (*p* < 0.05). “Family function” also has a statistically significant impact on “satisfaction with social support” (*p* < 0.05).

Then, the Bootstrap method was used to test whether “satisfaction with social support” is a mediating variable, and it was found that the direct effect is statistically significant (the confidence interval does not include 0), indicating that “satisfaction with social support” affects “loneliness” directly; indirectly, the mediating effect is statistically significant (the confidence interval does not include 0), indicating that “social support satisfaction” is the mediating factor between “family function” and “loneliness.”

## 4. Discussion

Empirical studies have shown that digital health interventions (DHIs) may enable low-cost, scalable improvements in the quality of care for adults with schizophrenia [34]. Healthcare consultants can apply AI mind maps to empower patients with schizophrenia to think about how to mobilize their social support more effectively. Relevant studies have indicated that social supports determine a caregiver’s care burden [13,35]; it has also been proven that good social support for patients—including the feeling of being cared for and respected by others—can reduce social discrimination, improves mental health, and correlates positively with recovery [13,36]. The World Health Organization’s (WHO) proposal in the 2013–2030 integrated mental health action plan regarding the need to provide holistic, comprehensive mental health social care services in community settings to support the community life of patients with schizophrenia [37]. Successful social reintegration can help to combat the stigma associated with schizophrenia. When individuals actively participate in their communities, it fosters understanding and acceptance among the public. This not only benefits those with schizophrenia but also enriches the community by promoting diversity and inclusion. Efforts to reintegrate individuals into society can lead to a more supportive environment that recognizes the capabilities and contributions of people with mental disabilities.

Although this study is innovative in exploring the application of AI mind maps in health education for patients with schizophrenia, it has multiple methodological limitations. The most important problem is the lack of control group design and randomization, which seriously limits the ability to draw causal inferences. In addition, problems such as small sample size, single institutional source, and lack of standardization of intervention methods also affect the credibility and generalizability of the research results. Future studies need to adopt more rigorous experimental designs, especially randomized controlled trials, to better evaluate the empowering effect of AI mind map intervention.

Rawat et al.’s visionary study showed that more studies are needed to understand the generative capabilities of artificial intelligence systems, and the specific context of LLMs is essential for researchers, practitioners, and policymakers to collaborate in shaping the responsible and ethical integration of these technologies in various fields in future [38]. The areas where AI mind map application research can be expanded include the following: (1) Expanding the application of AI mind mapping to other mental illnesses, such as depression, anxiety, bipolar disorder, etc. (2) Developing AI mind mapping tools suitable for children and adolescents. (3) Studying the application of AI tools in elderly mental health. (4) In-depth studies of the mechanism of AI tools to improve cognitive function. (5) Expanding research into family caregiver needs and supports. (6) Establishing a standardized AI mind map intervention protocol.

## 5. Conclusions

Through rigorous survey design and statistical analysis, this study successfully verified the mediating role of social support satisfaction between family function and patients’ quality of life and loneliness, provided empirical support for the application of AI mind maps in health education for patients with schizophrenia, and laid the foundation for the development of an intelligent integrated mental health care model.

Future development will move towards a more precise, personalized, and integrated direction. Through the collaborative efforts of technological innovation, policy support, and social participation, digital tools such as AI mind maps are expected to become an important part of mental health care and ultimately realize a modern mental health care system that is patient-centered, evidence-based, and technology-supported.

This is not only a technological advancement, but also an innovation in the concept of care, which will bring better quality of life to patients with mental disabilities, create greater value for society, and make important contributions to the development of human mental health.

## Figures and Tables

**Figure 1 healthcare-13-01885-f001:**
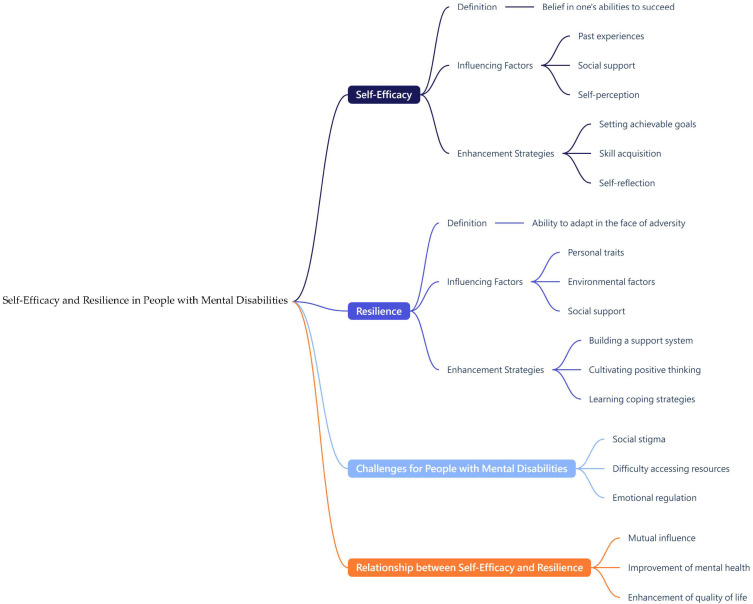
An AI Mind Map of Self-Efficacy and Resilience in People with Mental Disabilities.

**Figure 2 healthcare-13-01885-f002:**
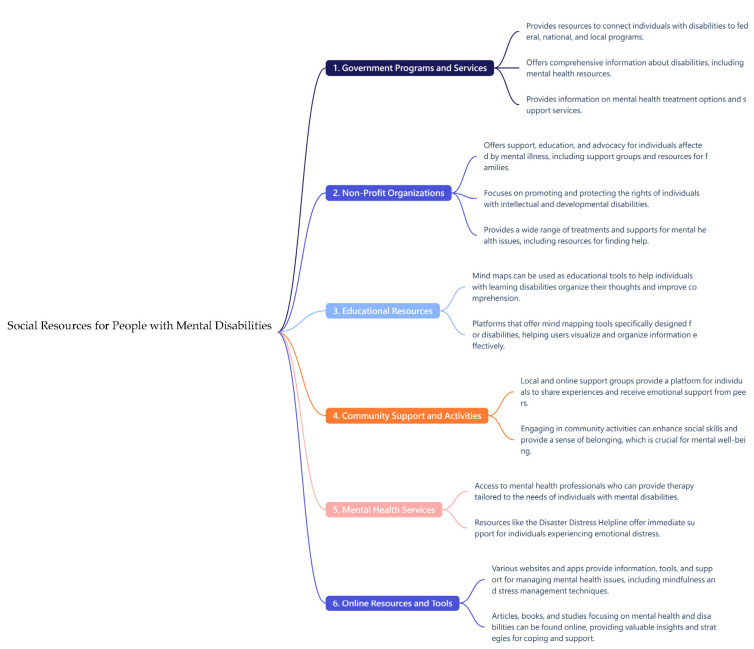
An AI Mind Map of Social Resources for People with Mental Disabilities.

**Figure 3 healthcare-13-01885-f003:**
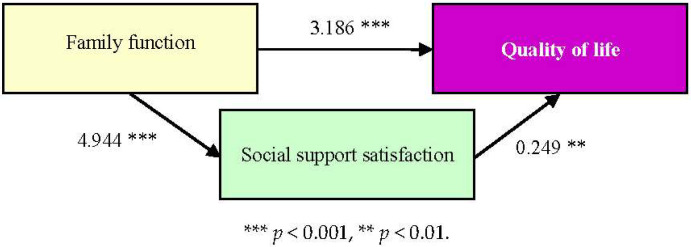
Relationship between family function, social support satisfaction (as a mediating factor), and quality of life.

**Figure 4 healthcare-13-01885-f004:**
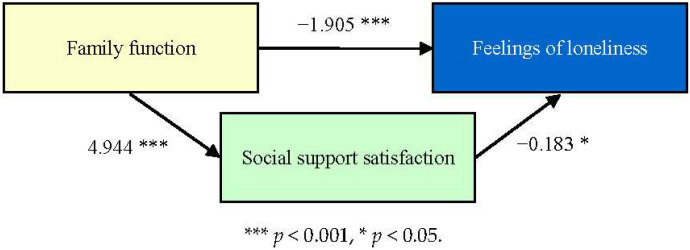
Relationship between family function, social support satisfaction (as a mediating factor), and loneliness.

**Table 1 healthcare-13-01885-t001:** Analysis of social support satisfaction as a mediating variable between family function and quality of life.

Variables	β	SE	95%C.I.		Bootstrap 95%C.I.	
			Lower	Upper	Lower	Upper
Family function→Quality of life	3.186 ***	0.568	2.047	4.324	2.047	4.406
Family function→Social support	4.944 ***	0.752	3.438	6.451	3.470	6.359
Family function→Social support→Quality of life	0.249 **	0.076	0.096	0.402	0.123	0.387
	**Effect**	**SE**	**95%C.I.**		**Bootstrap 95%C.I.**	
Total effect	4.418 ***	0.461	3.494	5.342	─	─
Direct effect						
Family function→Quality of life	3.186 ***	0.568	2.047	4.324	─	─
Indirect effect						
Family function→Social support→Quality of life	1.232 ***	0.332	─	─	0.627	1.931

*** *p* < 0.001, ** *p* < 0.01.

**Table 2 healthcare-13-01885-t002:** Analysis of social support satisfaction as a mediating variable between family function and loneliness.

Variables	β	SE	95%C.I.		Bootstrap 95%C.I.	
			Lower	Upper	Lower	Upper
Family function→Feelings of loneliness	−1.905 ***	0.520	−2.947	−0.862	−2.784	−0.981
Family function→Social support	4.944 ***	0.752	3.438	6.451	3.470	6.330
Family function→Social support→Feelings of loneliness	−0.183 *	0.070	−0.323	−0.043	−0.302	−0.071
	**Effect**	**SE**	**95%C.I.**		**Bootstrap 95%C.I.**	
Total effect	−2.811 ***	0.409	−3.632	−1.991	─	─
Direct effect						
Family function→Feelings of loneliness	−1.905 ***	0.520	−2.947	−0.862	─	─
Indirect effect						
Family function→Social support→Feelings of loneliness	−0.906 **	0.313	─	─	−1.571	−0.337

*** *p* < 0.001, ** *p* < 0.01, * *p* < 0.05.

## Data Availability

All of the relevant datasets in this study are described in the manuscript.

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
