# Peer review of "Application of AI Mind Mapping in Mental Health Care"

_healthcare, 2025, doi:10.3390/healthcare13151885_

Round 1
Reviewer 1 Report
Comments and Suggestions for Authors
- The study uses regression and mediation analysis to explore relationships between family function, social support, and patient outcomes, however, the absence of a control or comparison group, for example, a group not exposed to AI mind mapping, limits the ability to attribute observed improvements directly to the intervention.
-
The manuscript references the use of "Monica AI" for mind mapping but does not provide sufficient detail on the AI model’s underlying architecture, decision logic, or level of automation. Is it rule-based or learning-based?
-
Although the study uses a purposive sampling method, more discussion is needed on how the sample size (n=66) was justified statistically and to what extent the findings can be generalized beyond the selected hospital setting in Taiwan.
-
The mediation analysis is statistically sound, but the clinical significance of the effect sizes is not well-discussed. For example, how meaningful is a 1.23-point indirect effect in practical terms for a patient’s quality of life?
-
The manuscript briefly links the findings to WHO recommendations, which is commendable. However, the discussion could benefit from integrating more global perspectives on digital tools in mental health care, especially in light of growing interest in AI-based therapeutic aids. Consider adding a brief comparison with existing digital mental health interventions would provide valuable context.
- Check the format, fonts size is incorrect as per the journal template.
Author Response
Application of AI Mind Mapping in Mental Health Care
Response to Reviewer A Comments
Comments 1,
The study uses regression and mediation analysis to explore relationships between family function, social support, and patient outcomes, however, the absence of a control or comparison group, for example, a group not exposed to AI mind mapping, limits the ability to attribute observed improvements directly to the intervention.
Response 1,
Thanks to the reviewer A for valuable comments. We have combined with valuable suggestions in the discussion (Line 321-325).
Comments 2,
The manuscript references the use of "Monica AI" for mind mapping but does not provide sufficient detail on the AI model’s underlying architecture, decision logic, or level of automation. Is it rule-based or learning-based?
Response 2,
Monica AI can be considered more of a decision logic function rather than an automation function. Here’s a brief breakdown:
Decision Logic Function
- Mind Mapping: Helps in organizing thoughts, ideas, and decision-making processes.
- Analysis: Assists in evaluating options and outcomes based on user inputs.
Automation Function
- Task Execution: Involves performing tasks automatically without human intervention.
- Process Automation: Typically requires predefined rules and workflows.
Comments 3,
Although the study uses a purposive sampling method, more discussion is needed on how the sample size (n=66) was justified statistically and to what extent the findings can be generalized beyond the selected hospital setting in Taiwan.
Response 3,
This study calculated the sample size using G*power version 3.1 statistical soft-ware, with an effect size of 0.15, an α error probability of 0.05, and an estimated sample size are 52 persons with schizophrenia. Because it is difficult for persons with schizophrenia to see an outpatient regularly, the estimated sample size was increased by 20% due to the loss rate. Research limitations are mentioned at the conclusion of this study: This study was an uncontrolled, cross-sectional, purposive sampling survey study, so the findings may not be generalizable to other groups. In the future, it is worth expanding the sample size and conducting randomized controlled trials to verify the effectiveness of AI mind mapping in empowering patients.
Comments 4,
The mediation analysis is statistically sound, but the clinical significance of the effect sizes is not well-discussed. For example, how meaningful is a 1.23-point indirect effect in practical terms for a patient’s quality of life?
Response 4,
"Social support satisfaction" affects "quality of life"; the mediating effect of the indirect effect is statistically significant (the confidence interval does not include 0), indicating that "social support satisfaction" is a mediating factor between "family function" and "quality of life" (it is a partial mediator).
Comments 5,
The manuscript briefly links the findings to WHO recommendations, which is commendable. However, the discussion could benefit from integrating more global perspectives on digital tools in mental health care, especially in light of growing interest in AI-based therapeutic aids. Consider adding a brief comparison with existing digital mental health interventions would provide valuable context.
Response 5,
Empirical studies have shown that Digital health interventions (DHIs) may enable low cost, scalable improvements in the quality of care for adults with schizophrenia [37].
Comments 6,
Check the format, fonts size is incorrect as per the journal template.
Response 6,
We check the format, fonts size again to meet the journal template requirements.
Reviewer 2 Report
Comments and Suggestions for Authors
Comments
Overall
The perspective of social reintegration support in patients with schizophrenia is a very important perspective. In some countries, the illnesses prevent patients from reintegrating into society, and in many cases, they are forced to receive public assistance because their workplaces are limited or it is difficult for them to work. Furthermore, the number of people affected by these diseases has been on the rise in recent years, and some say that the multimedia environment, such as SNS, is the cause of this increase, so countermeasures are urgently needed.
(1) The issue in this paper is whether it is important to initially set the intensity of tasks that schizophrenia sufferers can perform and make normal judgments about. Also, how much can we ask an AI to do and how much can we get an accurate answer? It is also important to create a template or have the AI understand what it can ask and to what extent it can provide accurate answers.
It is also important to have the patient understand that the AI is a “treatment support tool” so that the patient does not have any suspicion toward the generated AI.
The mind map in Figure 1 should be pasted in the introduction to make it easier for the reader to decipher the previous studies.
(2) In the present study, we focused on the family structure and other factors, but not on the cognitive abilities of the subjects and the range of their initial abilities. Do you plan to make use of this in future studies and surveys?
We would like to see those discussions.
(3) Differences from previous studies
The previous studies in this paper refer to relatively old literature, but it would be helpful to have more suggestions and references to the use of generative AI and LLMs in presenting the approach in this paper to guarantee the discussion.
(4) Significance of social reintegration in schizophrenia
The above should be the most important perspective in this paper. This should be emphasized a little more in the introduction.
(5) Conclusion
I think we should discuss some more negative issues of the discussion based on the suggestions of the comments.
Author Response
Application of AI Mind Mapping in Mental Health Care
Response to Reviewer B Comments
Overall
The perspective of social reintegration support in patients with schizophrenia is a very important perspective. In some countries, the illnesses prevent patients from reintegrating into society, and in many cases, they are forced to receive public assistance because their workplaces are limited or it is difficult for them to work. Furthermore, the number of people affected by these diseases has been on the rise in recent years, and some say that the multimedia environment, such as SNS, is the cause of this increase, so countermeasures are urgently needed.
Comments 1,
The issue in this paper is whether it is important to initially set the intensity of tasks that schizophrenia sufferers can perform and make normal judgments about. Also, how much can we ask an AI to do and how much can we get an accurate answer? It is also important to create a template or have the AI understand what it can ask and to what extent it can provide accurate answers.
It is also important to have the patient understand that the AI is a “treatment support tool” so that the patient does not have any suspicion toward the generated AI.
The mind map in Figure 1 should be pasted in the introduction to make it easier for the reader to decipher the previous studies.
Response 1,
Thanks to the reviewer B for valuable comments, the literature on AI or digital mental health interventions is diverse, highlighting their potential benefits and challenges. While they offer promising solutions for enhancing mental health care accessibility, ongoing research is necessary to address limitations and optimize their effectiveness.
The AI mind map in Figure 1 had be adjusted pasted in the introduction to make it easier for the reader to decipher the previous studies (Line 103-106).
Comments 2,
In the present study, we focused on the family structure and other factors, but not on the cognitive abilities of the subjects and the range of their initial abilities. Do you plan to make use of this in future studies and surveys?
We would like to see those discussions.
Response 2,In the conclusion, the researchers mentioned how future research can overcome the limitations of this study: Healthcare consultants can apply AI mind maps to empower patients with schizophrenia to think about how to mobilize social supports more effectively. This study was an uncontrolled, cross-sectional, purposive sampling survey study, so the findings may not be generalizable to other groups. In the future, it is worth expanding the sample size and conducting randomized controlled trials to verify the effectiveness of AI mind mapping in empowering patients.
Comments 3,
Differences from previous studies
The previous studies in this paper refer to relatively old literature, but it would be helpful to have more suggestions and references to the use of generative AI and LLMs in presenting the approach in this paper to guarantee the discussion.
Response 3,
This study complements 2024 empirical studies that have shown that Digital health interventions (DHIs) may enable low cost, scalable improvements in the quality of care for adults with schizophrenia [37].
Comments 4,
Significance of social reintegration in schizophrenia
The above should be the most important perspective in this paper. This should be emphasized a little more in the introduction.
Response 4,
Thanks to the reviewer's reminder, already added in the introduction, social reintegration for individuals with schizophrenia is a critical aspect of their recovery and overall well-being. This process involves helping individuals with schizophrenia return to their communities and engage in social, educational, and occupational activities (Line36-39).
Comments 5,
Conclusion
Application of AI Mind Mapping in Mental Health Care
Response to Reviewer B Comments
Overall
The perspective of social reintegration support in patients with schizophrenia is a very important perspective. In some countries, the illnesses prevent patients from reintegrating into society, and in many cases, they are forced to receive public assistance because their workplaces are limited or it is difficult for them to work. Furthermore, the number of people affected by these diseases has been on the rise in recent years, and some say that the multimedia environment, such as SNS, is the cause of this increase, so countermeasures are urgently needed.
Comments 1,
The issue in this paper is whether it is important to initially set the intensity of tasks that schizophrenia sufferers can perform and make normal judgments about. Also, how much can we ask an AI to do and how much can we get an accurate answer? It is also important to create a template or have the AI understand what it can ask and to what extent it can provide accurate answers.
It is also important to have the patient understand that the AI is a “treatment support tool” so that the patient does not have any suspicion toward the generated AI.
The mind map in Figure 1 should be pasted in the introduction to make it easier for the reader to decipher the previous studies.
Response 1,
Thanks to the reviewer B for valuable comments, the literature on AI or digital mental health interventions is diverse, highlighting their potential benefits and challenges. While they offer promising solutions for enhancing mental health care accessibility, ongoing research is necessary to address limitations and optimize their effectiveness.
The AI mind map in Figure 1 had be adjusted pasted in the introduction to make it easier for the reader to decipher the previous studies (Line 103-106).
Comments 2,
In the present study, we focused on the family structure and other factors, but not on the cognitive abilities of the subjects and the range of their initial abilities. Do you plan to make use of this in future studies and surveys?
We would like to see those discussions.
Response 2,In the conclusion, the researchers mentioned how future research can overcome the limitations of this study: Healthcare consultants can apply AI mind maps to empower patients with schizophrenia to think about how to mobilize social supports more effectively. This study was an uncontrolled, cross-sectional, purposive sampling survey study, so the findings may not be generalizable to other groups. In the future, it is worth expanding the sample size and conducting randomized controlled trials to verify the effectiveness of AI mind mapping in empowering patients.
Comments 3,
Differences from previous studies
The previous studies in this paper refer to relatively old literature, but it would be helpful to have more suggestions and references to the use of generative AI and LLMs in presenting the approach in this paper to guarantee the discussion.
Response 3,
This study complements 2024 empirical studies that have shown that Digital health interventions (DHIs) may enable low cost, scalable improvements in the quality of care for adults with schizophrenia [37].
Comments 4,
Significance of social reintegration in schizophrenia
The above should be the most important perspective in this paper. This should be emphasized a little more in the introduction.
Response 4,
Thanks to the reviewer's reminder, already added in the introduction, social reintegration for individuals with schizophrenia is a critical aspect of their recovery and overall well-being. This process involves helping individuals with schizophrenia return to their communities and engage in social, educational, and occupational activities (Line36-39).
Comments 5,
Conclusion
I think we should discuss some more negative issues of the discussion based on the suggestions of the comments.
Response 5,
I am very grateful for the reviewers' insights. The limitations and improvement methods of this study have been mentioned in the discussion and conclusions (Line321-325; Line348-352). The negative impact of lack of social integration on mental health of people with schizophrenia has also been mentioned in the conclusions (Line342-348).
Round 2
Reviewer 2 Report
Comments and Suggestions for Authors
Overall
Thank you for reflecting my previous comment. On that basis, I have a further question.
(1) You mentioned that you recruited patients with schizophrenia, did you conduct a positive response to those surveys, including compensation for participation, and did you scrutinize them for bias?
Please be more specific about the survey design, whether you did this as a hospital initiative or not.
How did you provide care and reasonable accommodations for patients undergoing treatment? Depending on the symptoms, it is quite possible to assume incoherent answers. I was also concerned that there was no discussion of adjustments in response design or review of previous studies.
Please be specific in your description of the above.
Author Response
Application of AI Mind Mapping in Mental Health Care
Response to Reviewer Round 2 Comments
Overall
Thank you for reflecting my previous comment. On that basis, I have a further question.
Comments 1,
(1) You mentioned that you recruited patients with schizophrenia, did you conduct a positive response to those surveys, including compensation for participation, and did you scrutinize them for bias?
Please be more specific about the survey design, whether you did this as a hospital initiative or not.
How did you provide care and reasonable accommodations for patients undergoing treatment? Depending on the symptoms, it is quite possible to assume incoherent answers. I was also concerned that there was no discussion of adjustments in response design or review of previous studies.
Please be specific in your description of the above.
Response 1,
Thanks to the reviewer Round 2 for valuable comments.
- The researcher reflected in the discussion. The study uses regression and mediation analysis to explore relationships between family function, social support, and patient outcomes, however, the absence of a control or comparison group, for example, a group not exposed to AI mind mapping, limits the ability to attribute observed improvements directly to the intervention.
- At the same time, the researchers also mentioned the limitations of this study in the conclusion, and methods that can be improved in future studies, such as this study was an uncontrolled, cross-sectional, purposive sampling survey study, so the findings may not be generalizable to other groups. In the future, it is worth expanding the sample size and conducting randomized controlled trials to verify the effectiveness of AI mind mapping in empowering patients.
- The participants of this study were patients with schizophrenia who attended the psychiatric outpatient clinic of a hospital and all lived at home in the community.
- All outpatient study participants must have been assessed by their psychiatrist as having stable mental symptoms.
- Finally, the research cited by Dr. Rawat et al.'s visionary study, which showed that more studies are needed to understand the generative capabilities of artificial intelligence systems, and the specific context of LLM is essential for researchers, practitioners, and policymakers to collaborate in shaping the responsible and ethical integration of these technologies in various fields as the direction of efforts [38].
Thank you very much for your careful review, which has taught me a lot. Thank you!
Round 3
Reviewer 2 Report
Comments and Suggestions for Authors
Basically, the points I made last time are the same, but I will be a little more specific.
(1) The issue in this paper is whether it is important to first set the intensity of the tasks that schizophrenia patients can perform and make normal judgments. It is also necessary to consider how much can be asked of the AI ​​and how accurate the answers can be obtained. It is also important to create a template for the AI ​​to understand what it can be asked and how accurate the answers it can provide. It is also important for patients to understand the AI ​​as a "treatment support tool" and not to have doubts about the generated AI. The mind map in Figure 1 should be pasted in the introduction to make it easier for readers to decipher past research.
(2) Lack of description of the specific operation of AI mind maps
The provided section of the paper mentions the purpose and situations in which AI mind maps were used, such as "AI mind maps were used during regular outpatient clinic attendance for individual and group health education," "healthcare consultants can apply AI mind maps to empower patients with schizophrenia to think about how to mobilize their social support more effectively," and "explore healthcare consultants' application of AI mind maps to educate patients with schizophrenia in their perceptions of their family function, social support, quality of life, and loneliness, as well as empowering these patients to think more organizationally and understand problems more effectively."
However, as the review comments point out, there are no detailed descriptions of the specific design, operation protocol, quality control, and introduction method of the AI ​​mind map, such as how the "intensity of the task" performed by the patient was set and adjusted, how much the AI ​​could be requested to do (the scope of the AI's functions), how the "accuracy" of the answers provided by the AI ​​was guaranteed and evaluated, the presence and content of "templates" to support interaction with the AI, and specific approaches for patients to understand and trust the AI ​​as a "treatment support tool." These are essential information for evaluating the reproducibility of research and the quality of the intervention.
About the mind map in Figure 1:
The provision part of the paper does not include Figure 1 itself or an explanation of its contents. Therefore, it is not possible to determine what Figure 1 is and to what extent it contributes to the deciphering of past research. However, in general, showing related figures in the introduction is an effective way to help understand the background and framework of the research.
(2) Questions
A. How was the "task intensity" of the AI ​​mind map used in this study adjusted according to the stability of the patient's symptoms ( "stable mental symptoms") and individual differences in ability?
B. What kind of "requests" (questions, instructions, etc.) are expected to be made by the patient to the AI ​​mind map, and how was the "accuracy" of the AI's response to those requests verified in advance or monitored during the study?
C. Were "templates" or specific guidance provided to help patients effectively use the AI ​​mind map? If so, is it possible to specifically indicate the content of those?
D. What explanations or approaches did the medical consultant provide to help patients positively accept the AI ​​as a "treatment support tool" and avoid excessive expectations or unnecessary doubts? (e.g., clarifying the capabilities and limitations of the AI, explaining ethical considerations, etc.)
E. If Figure 1 is placed in the introduction of a paper, what specific contributions can be expected to make to readers' understanding of the theoretical background of this study and its relevance to previous research? (Since the entire paper has not been provided, we would like to ask the author for his intention.)
(3) This study focused on family structure and other factors, but not on the range of subjects' cognitive abilities or initial abilities. Are there plans to use this method in future studies or surveys? I would like to see that discussion.
In the paper, gender, age, educational background, and religious beliefs were collected as a "Demographic variable questionnaire," and these psychosocial factors were measured using the "Family function scale," "Social support scale," "Quality of life scale," and "Loneliness scale."
However, there is a lack of description of cognitive and initial abilities.
As eligibility criteria for participants, "assessed by their psychiatrist as having stable mental symptoms" and "literate, able to respond to the topic" are required to have a certain degree of communication ability and symptom stability, but there is no description of quantitatively evaluating and adding to the analysis the cognitive functions (memory, attention concentration, executive function, etc.) of schizophrenia patients or their initial skill level in using AI mind maps (for example, IT literacy, etc.).
(4) Questions
A. If you plan to measure and evaluate the subjects' "cognitive abilities" or "initial abilities" in your future research, what specific evaluation indicators or scales (e.g., BACS-J, CANTAB, your own IT skills assessment, etc.) are you considering using?
B. What is the hypothesis that these cognitive abilities and initial abilities are related to the experience of using AI mind maps and the main variables measured in this study, such as family functioning, social support, QOL, and loneliness?
C. If you measure these abilities, how do you plan to handle these variables in the research design and statistical analysis? (e.g., by including them as covariates, conducting subgroup analyses, incorporating them into models for mediation analysis or moderated effect analysis, etc.)
(5) Differences from past research Although the past research referenced in this paper is relatively old, when presenting the approach of this paper, it would be useful to strengthen the basis of the discussion by presenting more suggestions and references on the use of generative AI and LLM.
The citations in the provided excerpt are from the 1980s, such as Smilkstein et al. (1982) [26] (line 200), Sarason, Levine, Basham, and Sarason (1983) [27,28], and Russell et al. (1980) [32,33] (line 232), so the review comment that the literature is "relatively old" is valid.
The conclusion of the paper mentions the future importance of LLMs, stating, "The research cited by Rawat et al.'s visionary study showed that more studies are needed to understand the generative capabilities of artificial intelligence systems, and the specific context of LLMs is essential for researchers, practitioners, and policymakers to collaborate in shaping the responsible and ethical integration of these technologies in various fields in future [37]." The paper also cites references [37]. However, it is not clear from the provided portion what specific generative AI/LLM technology the "AI Mind Map" used in this study is based on, or to what extent it is based on the latest research trends related to them.
(6) Questions
A. What specific AI technology (specific LLM, unique algorithm, etc.) is the "AI Mind Map" used in this study based on? Can you explain the reason for selecting the technology and compare it with existing similar technologies?
B. Would it be possible to clarify the positioning and contribution of this study by adding more recent research papers and practice reports on the use of generative AI and LLM in supporting patients with schizophrenia (e.g., symptom management, cognitive rehabilitation, social skills training, therapeutic dialogue, etc.) as references?
C. If there is a background to referencing older literature (e.g., adoption of established scales or theoretical frameworks), would it be possible to clarify this point and discuss the novelty and progress of this study in terms of its use of AI in comparison with the latest research trends?
(7) The significance of social reintegration in schizophrenia The above should be the most important viewpoint in this paper. It should be emphasized somewhat in the introduction.
"Successful social reintegration can help to combat the stigma associated with schizophrenia. When individuals actively participate in their communities, it fosters understanding and acceptance among the general public. This not only benefits those with schizophrenia, but also enriches the community by promoting diversity and inclusion. Efforts to reintegrate individuals into society can lead to a more supportive environment that recognizes the capabilities and contributions of people with mental health conditions." The significance of social reintegration is described in relatively detail. However, as the review comment points out, it is not possible to judge from the information provided alone whether the importance of this theme is clearly conveyed to the reader in the introduction, which motivates the research and raises the issue. If the reference in the introduction is weak or abstract, there is room for improvement.
Questions
A. (Assuming that the entire paper has not been provided) In the introduction of this paper, how specifically are you discussing the challenges that schizophrenia patients face in reintegrating into society (e.g., stigma, difficulty in employment, social isolation, etc.) and the benefits that overcoming these challenges brings to individuals and society?
B. The review comment states that it "should be emphasized somewhat," but from your perspective, do you think that the significance of social reintegration is sufficiently emphasized in the current description of the introduction? If so, could you please specifically indicate which part and how it is emphasized?
C. Alternatively, wouldn't it be possible to more effectively appeal to the need for this research by presenting the multifaceted significance of social reintegration (reducing stigma, promoting diversity and inclusiveness in communities, etc.) as stated in the conclusion earlier and more forcefully in the introduction?
D. Regarding the use of AI mind maps proposed in this study, apart from the methodological limitations already mentioned in the discussion of the paper, are there any potential risks, ethical issues, or barriers to practical application that the authors recognize (e.g., privacy, bias, over-reliance, digital divide, information quality, etc., as mentioned above)?
E. Regarding these "negative aspects," what preventive measures, countermeasures, or issues that should be examined in future research do you think there are?
F. By specifically discussing these potential issues and their countermeasures in the conclusion or throughout the discussion, the authors could demonstrate their insights toward the responsible use of AI technology. This would be consistent with the perspective of "responsible and ethical integration of these technologies" as stated in Rawat et al. [37].
I hope that these discussions will reinforce the intention of your review comments and help you have a constructive dialogue with the paper authors.
